# Brazilian Environment and Plants as Seen by Japanese Eyes Two Hundred and Twenty Years Ago

**DOI:** 10.3390/plants13020188

**Published:** 2024-01-10

**Authors:** Natalia Hanazaki

**Affiliations:** 1Department of Environmental Sciences, Informatics and Statistics, Ca’Foscari University, Via Torino 155, Mestre, 30170 Venezia, Italy; natalia.hanazaki@ufsc.br; Tel.: +55-48-3721-9460; 2Department of Ecology and Zoology, Universidade Federal de Santa Catarina, Campus Universitário s/n, Florianópolis 88010-970, Brazil

**Keywords:** historical ethnobiology, historical ethnobotany, Atlantic Forest, navigation

## Abstract

In 2023, the Japanese migration to Brazil completed 115 years. However, the first time Japanese people arrived in Brazil and left a testimony of their experience was about two centuries ago. Their reports were registered in a historical document, handwritten during the Edo period when Japan was adopting a closed-door policy. The episode of their visit to Brazil is only a small part of the odyssey of these four Japanese sailors who departed from Ishinomiya to Tokyo at the end of the 18th century, but unexpectedly traveled around the globe. After a storm, they were adrift for six months until shipwrecking on the Aleutian Islands; from the Russian Aleutian Islands, they crossed the whole of Russia and boarded, in Saint Petersburg, on the first Russian expedition to circumnavigate the world. Their only stop in South America was at Santa Catarina Island, southern Brazil, and this is the first analysis of this episode from an ethnobiological perspective. Their reports described both the forest environment and the plants they observed and included at least 23 taxa of plants, mostly cultivated. These descriptions of plants and the environment are in contrast with other reports from the same period and to the current environment found in Santa Catarina Island, inspiring reflections on the construction of Brazil’s image in Japan before the 20th century.

## 1. Introduction

Being a Brazilian with Japanese ascendants whose grandparents migrated from Japan to Brazil at the beginning of the 20th century, the issue of migrations and identities always caught my attention. Why did my four grandparents, on different occasions, decide to move to the other side of the Earth more than a hundred years ago? Brazil and Japan are antipodes, or points diametrically opposite on the Earth, which means that to Japan, there is no place geographically farther than Brazil. Apart from the strong governmental propaganda promising plenty of land and resources in abundance for people suffering from hunger and other mishaps, people who decided to sail across the ocean knew practically nothing about the place they were choosing to live. Even with some governmental support, these deliberate migrations had their own costs because people had to adapt to the new country, with a different society and environment, facing prejudice, and after one or two generations, changing their mother tongue to Portuguese. The official registers of Japanese migrations to Brazil started at the beginning of the 20th century with the arrival of the ship Kasato Maru in 1908, preceded by bilateral negotiations since the end of the 19th century [1]. Since then, approximately 250 thousand Japanese people have arrived in Brazil as migrants [2], forming the largest Japanese population outside Japan with approximately two million descendants today [3].

To my astonishment, it was only in 2021 that I saw an internet post about the visit of the Japanese ambassador to Brazil to a memorial monument for the sailors of Wakamiya Maru [4]. This was when I first heard about this incredible story of four Japanese sailors who unwittingly traveled around the world more than a century before the massive Japanese migration to Brazil started. In addition, the one and only place in South America where they stopped was the exact city where I currently work and live. As I sought more information about this event, my surprise grew. This incredible story of these Japanese castaways, who crossed the whole Russian territory and, more than ten years after their shipwreck, returned to Japan aboard the first Russian expedition to circumnavigate the globe, departing from Saint Petersburg, is registered in a unique historical description: a manuscript from the Edo period, called *Kankai Ibun* (環海異聞, or exotic tales from) with 16 parts, first written ca. 1805 by Otsuki Gentaku and Shimura Hiroyuki [5].

Historical manuscripts such as these are rich sources for ethnobotanical investigations, even considering some imprecision. The oldest records (written in the usual languages of Western science) about the plants and the environment in Brazil are present in manuscripts written by European colonizers, which include documents such as Caminha’s letter [6]—analyzed from a botanical perspective by the authors of [7]—and other more detailed (Gandavo [8], first published in 1576) or later better-known descriptions, such as those after the passage of Piso [9] published in 1658, Humboldt and Bonpland [10] in 1820 and even Darwin [11] in 1832 (published 1859). Ethnobotany in historical documents is an approach widely explored in relation to medicinal and food plants in historical records in Brazil (e.g., [12]) and elsewhere in the world (e.g., [13,14]). The contribution of historical documents, both via written sources and iconography, provides valuable ethnobotanical information about the plants described [15], but also about the observers who contributed to science and sometimes remained suppressed in history, such as female naturalists [16]. The approaches to the information on historical documents can be guided by a search about a given species (in most cases) or themes, or even analyzing the contents of a document to infer about plants [17] and are not restricted to the plant scope (e.g., [18]), including important registers about landscape changes [19].

Although some authors explored parts of the amazing history enclosed in *Kankai Ibun*, including the material testimony and legacy in one jacket used by one of the sailors [20], the description of an Earth Globe witnessed in Saint Petersburg [21], and a theatrical event watched by the Japanese castaways when invited by Emperor Alexander I [22], no further analysis was performed under an ethnobiological perspective. In this article, I explore the testimonies in the written registers of *Kankai Ibun* about the months these Japanese pioneers spent anchored in Santa Catarina, Brazil, and compare these descriptions with other historical sources about the environment and the ecosystem of this region in this period between the 18th and 19th centuries.

The oral reports of the first Japanese to circumnavigate the world were materialized in several versions of the *Kankai Ibun* [5], a handwritten document, as mentioned above. Otsuki Gentaku, the main author of *Kankai Ibun*, was a *rangaku* (Dutch learning) scholar [23], and he was assisted by other scholars who wrote this manuscript based on other Western documents they had available at the time. They were careful to separate the information from the oral reports of the castaways and their own observations.

According to Ramming [21], “this work, now rather rare, was at the beginning of the 19th century widely circulated throughout Japan and most eagerly read by all interested in foreign things”. Such records often provide vivid descriptions of contemporary native societies with valuable historical, ethnographical, and anthropological data [24], and yet only a few scholars have analyzed these documents (mainly in Japanese publications but also in English, see [25]); their value remains much neglected simply as “curious tales” [24].

## 2. Results and Discussion

The odyssey of the castaways from Wakamiya Maru is described in several documents, both in Japanese [5] and summarized in English [21,26]. In Portuguese, in addition to Tomoko K. Gaudioso’s translation of volume 12 of *Kankai Ibun* [27] and her analysis of the first Japanese in Brazil [1], there are scattered newspaper notes and internet websites and a recent publication of a children’s book about this episode [28]. Nevertheless, since this is a story quite unknown to the general public, a brief summary is presented here.

At the end of the 18th century, in November 1793, the small ship Wakamiya Maru, with sixteen crew members, departed from Ishinomaki (currently Miyagi prefecture) to Edo (currently Tokyo), transporting 1300 bales of rice and 400 pieces of wood [29]. What was planned as a trip of a few weeks took an unexpected direction: after a strong storm near the coast of Fukushima, the ship went adrift for over 165 days, or more than five months, finally shipwrecking at the Aleutian Islands in the North Pacific Ocean. After one year in Unalaska, an Aleutian island under the Russian domain, the castaways were transferred to Irkutsk, in the middle of Siberia, where they spent another seven years. When the new Russian emperor Alexander I decided to support a Russian expedition with an embassy to establish commercial ties with Japan, the Japanese castaways were summoned to quickly travel to Saint Petersburg, where after an arduous journey of more than 40 days, they had an audience with the emperor. At this time, three crew members of Wakamiya Maru had already died (the captain Heibei, the crew member Kichirōji, and the sailor Ichigorō) and another three members decided to stay in Russia due to a weak health condition (the sailors Sadayū, Ginsaburō, and Saizō). Of the ten who attended the hearing with the emperor, four decided to return to Japan, and six opted to stay in Russia and be converted to Christianity (the sailors Taminosuke, Shigejirō, Yasaburō, Zenroku, Tatsuzō, and Minosuke) [29].

An important reminder is that during this period, Japan was under a policy of strict isolation [25], with the *sakoku seisaku* (鎖国政策) or “closed door policy” under the Tokugawa shogunate. This policy prohibited all foreign contact except for trade conducted officially with the Netherlands and China via Nagasaki port [24] and to some extent with Korea and the Ryūkyū kingdom (currently Okinawa). Thus, not only were foreign ships mostly forbidden to dock in Japan, but Japanese citizens were also forbidden to leave the country.

The four people who decided to return to Japan by joining the Russian expedition on board the main ship Nadezhda, were the cook of Wakamiya Maru, Gihei, born in Higashimatsushima, and the sailors Tsudayū, Sahei, and Tajuro. Tsudayū and Sahei were born in Shiogama and were approximately 59 and 41 years old, respectively, at the time the Russian expedition departed in August 1803, approximately ten years after they left Japan. Tajuro was the youngest, approximately 33 years old, and was born in the same village as Gihei (at that time, approximately 42 years old). Zenroku, who had already converted to Christianity and became a Japanese teacher in Irkutsk, also joined the expedition with the role of interpreter [29] and disembarked in Russia prior to their arrival back in Japan in 1804.

The Russian expedition first docked in Copenhagen, then at Falmouth (England) and Canaries Islands. After crossing the Atlantic Ocean and the equator line, in December 1803, the expedition stopped in only one port in South America, Ekaterina Island, or Island of Santa Catarina (currently Florianópolis and Governador Celso Ramos municipalities). There, they spent approximately 45 days to restock provisions and fix damages in one of the ships (Neva).

After departing from Santa Catarina Island on 4 February 1804, the expedition passed through Cape Horn, Easter Island, and Nuku Hiva (French Polynesia). In Hawaii, the ships parted ways: since the Russian expedition supposedly had only the permit for one ship to land in Nagasaki, the Nadezhda headed to the Kamchatka peninsula and then to Japan to attempt the diplomatic mission—which was unsuccessful at the end—and the Neva headed to the Pacific coast in North America.

The Russian ship Nadezhda reached the Japanese coast in Nagasaki at the beginning of October 1804, but only after six months of slow negotiations under the strict rules of the Japanese government; and after the diplomatic mission failed completely, the four castaways Tsudayū, Sahei, Gihei, and Tajuro were delivered to the Japanese officials in April 1805. This seems to have occurred just before the departure of Nadezhda.

The anonymity of the castaways is remarkable in the books registering this trip, authored by Krusenstern [30], Langsdorff [31], and Lisiansky [32], and published in 1813 and 1814. They are usually described as “the Japanese”. The commander of the expedition (Krusenstern) mentioned them as “the Japanese who in 1796 were captured in the Aleutic Islands” to be returned to Japan and to “secure the favor of the monarch [of Japan] and his ministers”, together with other valuable presents [30], reflecting the role of the four castaways in the diplomatic negotiations intended with Japan. When listing the crew and passengers of Nadezhda, he solely registered “five Japanese”, although he also described that “except MM. Horner, Tilesius, Langsdorff, and Laband, there were no foreigners on board either of the ships” [30]. They are mentioned by Krusenstern with words of disapproval, being “scarcely possible to imagine worse people than they were” (...) “lazy, dirty in their persons, always ill-humored”, with the exception of “an old man of sixty years of age” [30], probably referring to Tsudayū. According to Krusenstern [30], the relationships between Tsudayū’s group and Zenroku, the interpreter, were also hostile: “they lived in a continued state of warfare (...) because he [the interpreter] was rather more noticed by the ambassador than the others”. The Russian Nikolai Reazanoff was the ambassador in charge of the diplomatic mission with Japan. His disappointment with the failed task of establishing commercial relations with Japan likely contributed to the start of the conflict between Russia and Japan on the Kuriles island a few years after the passage of the ship Nadezhda in Nagasaki. One remarkable episode registered by both Krusenstern [30] and Langsdorff [31] was when Tajuro tried to kill himself on 16 January 1805, supposedly due to the frustration of having arrived in Japan but being unable to disembark.

After being left in the care of Japanese officials, they were sent to the capital Edo (currently Tokyo), and were submitted to 40 days of a detailed inquiry [21]. During this period, with the closed-door policy, Japanese castaways drifting to other countries faced several obstacles to returning home, such as the possibility of being arrested upon returning, and those who returned were subjected to a detailed investigation about the nature of their experiences abroad. To maintain the closed-door policy, repatriated castaways were not allowed to relate their experiences freely to ordinary citizens [24]. The written document produced from this inquiry, known as *Kankai Ibun* [5], contains the first reports of how Brazil was viewed by Japanese eyes in the very first years of the 19th century.

### Brazilian Environment and Plants through the Eyes of the Japanese Castaways

A disclaimer is necessary regarding this analysis through the eyes of the castaways: they reported what they saw to people who did not have experiences abroad and who interpolated their accounts with what was known in Japan during the Edo period. These accounts were transcribed many times and then translated. In other words, although the writers were careful to provide accurate descriptions, the original accounts from the castaways were filtered through these other lenses. These narratives were also burdened by the trauma of a trip lasting more than 10 years through unknown lands, by the uncertainty of returning to their home country, and by the uncertainty of knowing whether there would be freedom when they returned home. Additionally, Tajuro was the only one who disembarked whenever they had an opportunity to do so [26]. Tsudayū also disembarked in Santa Catarina [27]. However, after the attempt to end his own life months after his arrival at Nagasaki, Tajuro probably could not talk during the interrogatory. According to Oshima [26], after this incident, he never spoke again; although the captain of the Nadezhda reported that “after his wound had healed, he was frequently heard to say that the Russians were very good people, but himself a very bad man; and he wished his life might soon have an end” [30] (p. 282). Thus, it is possible that most accounts from the Brazilian land were probably told by Tajuro to their friends who stayed aboard most of the time and who then retold these accounts to their inquisitors.

Krusenstern’s [30] and Lisiansky’s [32] diaries registered the crossing of the equator on 16 November 1803. After that, they spotted the Brazilian coast in Cabo Frio (Rio de Janeiro state coast) on 12 December and anchored on Anhatomirim Island in Santa Catarina Bay on 21 December 1803. In *Kankai Ibun* [5], several misinterpretations appear both in relation to dates and to places, despite the efforts of Otsuki and Shimura to be as precise as possible. Among the numerous notes added by Otsuki and Shimura to the accounts they were registering, there are several pages describing the attempts to equip the Japanese calendar with the one used by the Russians. In one of these notes, they mention that they must have landed in *Ekaterina* or *Ecaterína* (エカテリナ, or Santa Catarina Island), *Burajiri* (ブラジリ, Brazil), having stayed there for more than 70 days. Such confusion also reflects the limited knowledge available during the Edo period in Japan about these faraway parts of the world. For example, Otsuki and Shimura supposed that they harbored in a place that could be southern to Santa Catarina Island, at the mouth of the Plata River (which is between Argentina and Uruguay), although they raised doubts about this based on the observation that Brazil was part of the Portuguese (ポルトガリ, *Porutogari*) territory [5].

After crossing the equator, the castaways knew they were heading toward South America and commented “To the south, from that point on, we no longer saw either the North Star or the constellation of Ursa Major, so we were very amazed and commented about it” [27]. In this section, they commented, “Sailing on the high seas, we spent two or three days in places where the color of the water was different. The water had a dark reddish color” [27]. This fact was remarkable, as it was also reported in [30] and was investigated by the naturalist Langsdorff [31], who concluded that “this effect arose from an immense number of little crabs which floated upon the surface of the water”. This phenomenon is caused by the final larval stage (Megalopa) of decapod crustaceans from the deep sea, which was described, for example, for *Moreiradromia antillensis* on the shelf break on the central coast of Brazil in 2002 [33].

One episode involving Japanese travelers aboard Nadezhda in South Atlantic waters was mentioned by Krusenstern [30]: “We caught but one shark, part of which was eaten, although it was not as good as a bonito. Our Japanese, however, ate the head and seemed to relish it much”. Large sharks were much more abundant before the commercial exploitation of the South Atlantic seas [34].

As the expedition approached the Brazilian coast, the increased heat was perceived in their observations about the need for daily baths, sometimes two or three times a day [27]. Apart from the misregistered dates (probably due to the different calendars used at that time [27]), they reported that in November 1803, they docked in a large port called *Ekaterina* (エカテリナ), supposed to be one of the largest ports in South America. This observation reveals the scattered knowledge they had about the places they were passing through, both for the Japanese travelers and for the transcribers, Otsuki and Shimura. At the beginning of the 19th century, Santa Catarina was far from being an important port in Brazil when compared to other ports such as Salvador and Rio de Janeiro, well established since the 16th century (at the end of the 17th century, Santa Catarina Island was marked simply as “sterile and desert” in Coronelli’s world globes). For Krusenstern, the choice for this port to restock provisions was due to the lowest fees when compared to major ports [30], cued by La Perouse [35], who navigated through these waters a few years before and mentioned the preference for Santa Catarina port instead of Rio de Janeiro to avoid the formalities of the large cities. Other chroniclers passing through Santa Catarina in the 18th century also commented on the convenience of this port due to the availability of wood and provisions for ships [36,37,38].

Then, they describe that the harbor was large but formed a very shallow bay, so large ships could not approach the shore. They also reported several small rivers flowing into this bay. These descriptions, although somewhat vague, reveal their perception of the North Bay of Santa Catarina Island and the wisdom at that time that already recognized the limitations of navigation. The submerged relief of the North Bay is a shallow bay, with an average depth of 5 m (in fact, this is a semiconfined system, as there is a connection with the South Bay in a constriction approximately 500 m wide, with greater depth, as well as its open connection to the ocean) [39].

A precious description of canoes is also found: “the natives’ boats were as thin and long as bamboo leaves. Its bottom consisted of a board nailed to a tree trunk split in a half” [27]. Single-trunk canoe carving is a traditional vessel in many parts of the Brazilian coast, including the Santa Catarina coast. Several chronists and travelers observed and described these single-trunk canoes on Santa Catarina Island, some of them large enough to transport 50 people [38]. According to Souza-Sobrinho [40], at that time, these canoes were made of woods such as *Aiouea glaziovii* (Mez) R.Rohde, *Terminalia kleinii* (Exell) Gere and Boatwr., *Cedrela fissilis* Vell., *Ficus organensis* Miq., *F. christianii* Carauta, *F. insipida* Willd., and *Schizolobium parahyba* (Vell.) S.F.Blake. Building and maintaining single-trunk dugout canoes is an activity still found currently in this area [41], with registers of the contemporary use of *C. fissilis*, *Ficus* sp., *S. parahyba,* and species of *Nectandra* and *Ocotea*. De Paula et al. [42] found canoes about 200 years old in this same region; thus, some canoes currently found in this area date back to the beginning of the 19th century and are still in use. In other words, the oldest canoes currently found on the Santa Catarina coast could be contemporary to those witnessed by the Japanese castaways.

According to *Kankai Ibun*, Tajuro disembarked and observed a village with approximately a thousand houses. He probably visited the village of São Miguel, where the ships were restocked with potable water. The distance from the port as described in *Kankai Ibun* is probably misregistered (20 ri, an old Japanese measure of distance, which would be equivalent to approximately 80 km) because the distance to São Miguel village is no more than 8 km from where they were docked. Tajuro and his colleagues interpreted the places based on the knowledge they had at that time, such as describing the roof tiles as being made of cherry tree bark—a species absent in this region—which is reliable in accuracy only regarding the color of the traditional ceramic roof tiles when seen from a distance. Ceramic clay tiles with a frustum-conical shape have been used since Portuguese arrival in Brazil in the 16th century [43].

According to *Kankai Ibun* [27], they observed “an enormous number of trees on the hills. Among the familiar trees were bergamot and orange” (*Citrus* spp.). The abundance and quality of oranges, limes, and lemons at Santa Catarina Island is described by other travelers (e.g., [36,38]). Krusenstern, Lisiansky, and Langsdorff [30,31,32] all remarked in their registers on the luxuriant forests with an abundance of resources. These descriptions could be embedded in the romantic ideal of European naturalists and travelers of that time, who described picturesque and exotic landscapes but were also guided by their own subjectivity [44]. In *Kankai Ibun*, the luxuriant vegetation also attracted the attention of Japanese travelers. Indeed, these descriptions converge to the physiognomy of the original Atlantic Forest of Santa Catarina [45].

When the Russian expedition docked at Santa Catarina, provisions of staple foods, fruits, and wood from forests were reported as abundant, while few areas were cultivated. The registers of deforestation of the island started with the first reports of ships stopping at Santa Catarina Island for wood supply, dating back to 1526, when a Spanish expedition traveling to the Mollucas Islands stayed there for four months, the time needed to build a small vessel [46]. At that time, the use of the environment for cultivation was at a very small scale for manioc, corn, and cotton production by Carijo indigenous people [46]. The strategic location and the abundance of wood and supplies attracted the attention of navigators until the end of the 18th century and the beginning of the 19th century. Less than three decades after the Japanese visit to Santa Catarina, Duperrey [47] did not find an abundance of provisions and resources. According to Caruso [46], from 1748 onward, deforestation for agriculture increased with the settlement of Azorean migrants, and changes in forest cover were already perceivable in the first years of 1800 [46].

The availability of trees at that time was essential for the supply of wood for the construction of a new mast for Neva. Japanese travelers observed the arrival of the wood in the port, which they described as “a very hard wood with red and black parts mixed together. The Russians called it *karasunazeriwa*” (カラスナゼリワ, probably *krasnoye derevo* or *krasnaya drevesina*) or red wood. Also related to woods, they reported “They brought to the ship exotic woods that are light red in color and have spots the color of egg yolk. The day we came to Nagasaki, several people told us that it was red sandalwood” [27]. They also reported that they heard that a wood called *suô* 蘇枋 (スオウ) was native to the region, but they did not see this wood, which was interpreted by Gaudioso [27] as the word referring to sappanwood (*Caesalpinia sappan* L., with occurrence in Asia) but in this case would be the South American equivalent brazilwood [27] (*Paubrasilia echinata* (Lam.) Gagnon, H.C. Lima and G.P. Lewis).

The identification of wood based only on color is very imprecise, and the descriptions can refer to several different genera. Other travelers at the end of the 18th century reported the abundance of trees used as high-quality wood but with scattered mentions of their color or other characteristics. Wood species from that time that could be identified based on occurrence and on Portuguese names include *Cedrela fissilis*, *Ocotea odorifera* (Vell.) Rohwer, and *Ocotea* spp. [37,38] (species identifications were based on [40]), and some of them are currently very rare in this area (e.g., *Ocotea catharinensis* Mez. [40]). In regard to brazilwood, other travelers mentioned the presence of “the Brazilian wood used for dyes” in Santa Catarina at the end of the 18th century and the beginning of the 19th [38,48].

*Paubrasilia echinata* was a heavily exploited resource until the middle of the 19th century [49]. Historical maps from the 16th century roughly depicted the presence of this species in the region, although precise information about the geographical distribution in these cartographies is hampered for two reasons: first, the need to show the precise localization and geographical accidents of the discovered lands; and second, the choice of not showing the distribution of resources of economic interest with both political and strategic purposes [50]. The accepted distribution of brazilwood is the Atlantic Forest further north of Santa Catarina [51,52]; however, these scattered registers of travelers and naturalists of the 18th and 19th centuries [38,48] inspire thoughts about the unsure possibility of a broader, albeit rare, distribution of this species. 

Tsudayū also disembarked and observed a rice mill. Although we find few mentions of rice cultivation (*Oryza* sp.) on the island, other authors mentioned the rice mill powered by the aqueduct in São Miguel [47], in the continental area, close to the place where ships sought water supplies. Probably both Tajuro and Tsudayū could observe the village of São Miguel, but is uncertain if they disembarked in Nossa Senhora do Desterro as well. Lisiansky [32] described the abundance of cereals and staples and provided a list of provisions that could be purchased, including rice, wheat (*Triticum* sp.), corn (*Zea mays* L.), coffee (*Coffea arabica* L.), and manioc (*Manihot esculenta* Crantz). The castaways believed that people at Santa Catarina had corn as their main staple, and they observed the prohibition of eating rice [27]. However, what they described as the corn flour prepared with hot water to make a kind of glue was probably the first description of *beiju* or *tapioca* by Japanese eyes. The main staple at that time was manioc, usually prepared as flour mixed with hot water and eaten instead of bread [31]. For Langsdorff [31], corn and white bread were found only among the “very highest and richest of the people”.

The Japanese travelers reported in *Kankai Ibun* several local products, such as “cabbages, turnips (thin, with no change in flavor), horseradish (round), Chinese melons, melons, watermelons, pumpkins, cucumbers, grapes, pepper (the fruits are small and the pepper trees grow like trees), oranges, walnuts (small), apples, sugarcane (the thick ones were the diameter of a fist” [27]). Travelers’ reports since the 18th century mentioned the presence of a remarkable variety of oranges (*Citrus* spp.) [36,38,53], and melons, watermelons (Cucurbitaceae), sugarcane (*Saccharum officinarum* L.), and grapes (*Vitis vinifera* L.) [38,53]. Few products listed by Lisiansky as purchased by the expedition [32] were the same as described in *Kankai Ibun*: lemons (*Citrus* sp.), pumpkins (*Cucurbita* sp.), and bananas (*Musa* sp.). While some of these crops and fruits were expected to be present in Santa Catarina Island at that time, others could be reported as similar to fruits known in Japan.

Their description of coconuts (*Cocos nucifera* L.) is as follows: “There were some very large fruits. The outer shell was thick. When you removed it, you could see the very hard inner shell, part of which looked like a person’s face. Its interior was full of oily meat, sweet like nuts”. The enslaved people placed these fruits in a container and swam to the ship to sell them. They reported that when they tried these fruits, “we feel the freshness in our mouth and forget the intense heat, so we buy them and eat them several times” [27]. According to *Kankai Ibun* [27], Tsudayū had brought a shell-like object that he used as a water container, and the writers of *Kankai Ibun* concluded that it was indeed green coconut, or *kokkosu* (コッコス).

One of the richest descriptions is related to bananas (*Musa* sp.), as green plants that produce 20 to 30 bunches of long things, with three longitudinal edges, without seeds, which are green at the beginning but when mature turn yellow (Figure 1). They reported the harvesting of these fruits when green and taking two days to mature, being sweet and white in the interior, and considered it similar to akebi [27]. Akebi (*Akebia quinata* (Thunb. ex Houtt.) Decne.) is a fruit native to Japan with a sweet flavor. It is worth noting the slight changes in the representations of bananas in different versions of *Kankai Ibun* (Figure 1 [5,54,55,56,57,58]). These differences reveal the novelty in the knowledge about this fruit in 19th century Japan, probably mixed with the exotism of unknown lands and plants.

In addition to the plants among purchased products and goods, they also described the cultivation of cotton, which was “identical to that observed in Japan”, but with larger leaves [27]. Frézier [53] described in detail the presence of cotton on the island, which he referred to as *Gossypium arboreum* L. or *Xylon* sp. [53]. At that time, cotton was reported by other voyagers as one of the main crops [38], and probably cultivated by Carijo indigenous people prior to European colonists [46]. There is an observation added by the writers of *Kankai Ibun*, Otsuki and Shimura, about the possibility of these cotton plants surviving for more than one year due to the warm weather.

Finally, *Kankai Ibun* also mentions several animals witnessed by the Japanese castaways’ eyes. Among the domestic animals, they observed pigs, with tusks and fatty meat, and cattle, also with fatty meat; dogs similar to those in Japan, and tabby three-colored cats, similar to those in Japan but seemingly more ferocious [27]. They also mentioned the lack of fish and the availability of shrimp in abundance [27]. The lack of fish could be due to climatic conditions (usually with a lower productivity in summer, the period when they were docked at Santa Catarina), because other reports from the 18th century mention the abundance of fish [36,37,53], but this can also indicate the beginning of the decline of local fish stocks. One fish caught their attention, “with a square shell similar to that of a turtle”, and with a skin similar to pufferfish (Figure 1), probably referring to *Sphoeroides testudineus*.

Non-domesticated animals include descriptions of long-tailed monkeys and “little birds with a very beautiful color, blue and with red beaks and nasal holes. They sing doing *kiu kiu*. When someone sticks out their tongue, they suck it with their beak” [27], probably referring to hummingbirds. They also described what is supposed to be the Procyonidae *Nasua nasua*, with whitish-gray fur, a long snout and striped tail, and a bad smell. Some of these animals were raised on the ship, but several died during the trip; one probably survived until they reached the port of Kamchatka. Oshima [59] presents excerpts of the diaries of other tripulants of the Russian expedition, such as Ratmanov and Levenstein (レーベンシュタイン, probably Lowenstern) describing other aspects of the animals found in this part of the expedition, including an episode on board with captured monkeys and the Japanese travelers. 

At the end of the reports about the stay at Santa Catarina Island, the document presents a drawing (Figure 1) and a description of a caiman brought on board. It is described as a four-legged cub of an animal called *garukaruzeru* (ガルカルゼル). The animal had thick and dark skin, scales on its feet, and thorns on its tail. Its mouth was full of mismatched teeth, and above the eyes were things that looked like calluses. Each paw had three nails, and the Japanese castaways reported that they were told that “these calluses above the eyes turn into horns when they grow and that they live both in the sea and in the mountains and that they even hunt and devour men. Looking at the drawing of the dragon, we think it looks similar. We even commented that it was truly a baby dragon” [27]. Following the presented drawing, they concluded that it was a caiman, or crocodile (*kokojiru*, ココジル). They also reported that the animal was killed and preserved by the use of a white chemical substance, the removal of viscera and eyes, and the replacement of the eyes with spheres. Lisiansky [32] also reported this episode: “we caught a young one and sent it on board the Nadezhda to the naturalists, who have preserved the skin. Although this little monster was scarcely a yard long alligator”. *Caiman latirostris*, or broad-snouted caiman, is distributed in the eastern part of South America and is easily found in wetlands and mangroves on Santa Catarina Island [60] until present day.

## 3. Materials and Methods

There were more than 30 manuscript copies of *Kankai Ibun* circulating in Japan’s Edo period [25]. The full version of *Kankai Ibun* was also published in a printed version in Tokyo in 1986 [61] and in a collection of anecdotal stories of shipwrecked people under the title *Hyōryū kidan zenshu* [21]. In this essay, I used Tomoko Kimura Gaudioso’s [27] Brazilian Portuguese translation of volume 12 of *Kankai Ibun* as a main textual source. Other versions were accessed and consulted for both text and images [5,54,55,56,57,58,61].

To triangulate information and to understand the context of the voyage at the end of the 18th and beginning of 19th centuries, I searched for historical documents with descriptions of the Russian expedition of Nadezhda and Neva ships and other documents describing the socioeconomic and environmental context of Santa Catarina Island (several of them compiled by Haro [62]). The main sources about the Russian circumnavigation used here were the reports of the Estonian captain of the expedition and commander of the main ship, Nadezhda, Adam Johann von Krusenstern [30], originally published in German in 1803; the diary of the Ukrainian commander of Neva, the second ship in the Russian expedition, Yuri Fyodorovich Lisiansky [32]; and the registers of one of the naturalists on board, the German doctor Georg Heinrich Freiherr von Langsdorff [31]. Ten years later, Langsdorff would return to Brazil as the consul general of Russia, in Rio de Janeiro, and after that, he would lead important naturalist expeditions in Brazil. Before joining the Russian circumnavigation expedition, Langsdorff practiced medicine in Lisbon, and thus was familiar with the idiom used in Brazil. His 1813 book includes one of the first written transcriptions of music in Santa Catarina, a “Brazilian Aria”. Souza-Sobrinho [40] mentioned that Langsdorff collected more than 80 species of wood on his first travel to Brazil; unfortunately, Langsdorff described the loss of most of the botanical samples due to humidity [31] and published only descriptions of pteridophytes collected on this trip [63], but none of them could be related to the Japanese descriptions. Other written reports from the Russian circumnavigation, such as those registered by Wilhelm Gottlieb Tilesius von Tilenau, the main naturalist on board Nadezhda, were not used due to the lack of contextual data to allow relations to *Kankai Ibun* registers. 

Botanical names mentioned in historical documents or in the literature were updated following Plants of the World Online [64] and World Flora Online [65].

## 4. Conclusions

Little is known about what happened to the four castaways after the interrogation in Edo. They supposedly returned back to their original towns, with the express prohibition to tell what they witnessed outside Japan. Tajuro and Gihei died a short time after returning to their city, at 36 and 45 years old, respectively [29]. Tsudayū passed away at 70 years old, and Sahei passed away at 67 years old [29].

The reports of these four people’s experiences around the world reveal pieces of information of places, plants, and animals that existed only in the imagination of Japanese people, or were otherwise places and plants never thought of before. The wide circulation of several copies of *Kankai Ibun* during the Tokugawa Shogunate period played a role in the construction of Japanese views on Brazil through the eyes of these four people, which certainly contributed profoundly to the future implementation of the friendship treaty signed between Japan and Brazil [1] that ultimately resulted in the migratory waves of the beginning of 20th century. These impressions and views may have also had subtle consequences in the building of the several identities of Japanese descendants in Brazil [66].

These registers are invaluable historical reports that may be read and interpreted with caution, filtered by all rich and dramatic context surrounding these events. In addition to the caution already mentioned in analyzing records made by third parties of facts stored in the memories of travelers, these reports may have incorporated distortions over time due to transcriptions and along successive translations. However, several treasures are still hidden behind these narratives. This amazing history of four people who unwittingly traveled around the world and almost from one pole to another certainly has much more to be explored under ethnobiological and ethnobotanical lenses. Beyond the Santa Catarina passage, rich reports about all other points of passage are included in *Kankai Ibun*. This is still a rich and underutilized ethnographic document, showing us “a glimpse of the world as it appeared to common Japanese sailors” [20].

Finally, in addition to testimonies related to biodiversity, historical records show different perspectives on environments and the world. They also help to understand how identities are built from the past to the present, with so many connections in a multicultural world.

## Figures and Tables

**Figure 1 plants-13-00188-f001:**
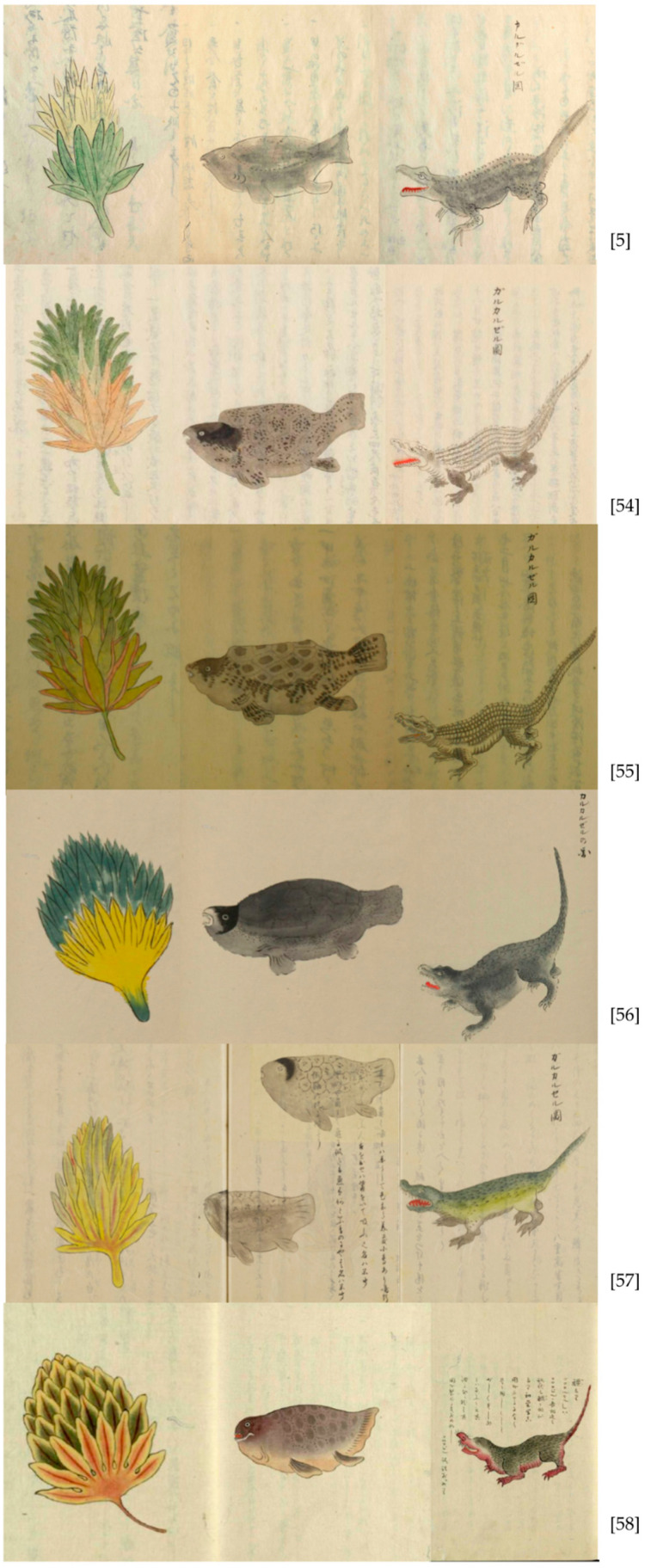
Examples of different versions of the drawings in *Kankai Ibun* on the nature elements observed in Brazil. From left to right: bananas (*Musa* sp.), pufferfish (probably *Spheroides testudineus*) and caiman (*Caiman latirostris*). The sources are indicated with the [5,54,55,56,57,58] in the bottom right of each series of pictures.

## Data Availability

Data is contained within the article.

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
