# Peer review of "Brazilian Environment and Plants as Seen by Japanese Eyes Two Hundred and Twenty Years Ago"

_plants, 2024, doi:10.3390/plants13020188_

Round 1

Reviewer 1 Report

Comments and Suggestions for Authors

See the suggestions for corrections and other recommendations in the attached file.

Comments on the Quality of English Language

See the suggestions for corrections and other recommendations in the attached file.

Author Response

Thanks for the quick review and suggestions provided. I accepted most of the suggestions, as explained in the response to each reviewer. All corrections were highlighted, except for the reference numbering, which were all changed from reference number 6 to 66 due to the inclusion of new citations.

Responses to Reviewer 1: 

All minor corrections (in the commented pdf) were accepted. Thank you very much for the careful review. I added a reference to Gandavo and other authors (as requested by Reviewer 3) to better position the documents analyzed here, among other historical and early scientific descriptions of Brazilian flora and environment.

The organization of sections follows the journal's guidelines, which is why the suggested changes regarding this issue were not accepted.

Reviewer 2 Report

Comments and Suggestions for Authors

Thank you for the opportunity to review this interesting paper. I found it original and well-written in the style of an ethnographic text that comes to illuminate details of the formation of people's relationship with the natural environment, especially that of exotic "other" cultures. I think it is useful to publish such articles, I found the observations on the species and uses of trees and changes in the landscape, the animals described etc. very interesting and valuable. I have not been able to check all the correspondences of official and scientific names, as most are unfamiliar to me. I do have one observation regarding the attribution of Caiman latirostris as a crocodile. The correct name is caiman.

Author Response

Thanks for the quick review and suggestions provided. I accepted most of the suggestions, as explained in the response to each reviewer. All corrections were highlighted, except for the reference numbering, which were all changed from reference number 6 until 66 due to the inclusion of new citations.

Response to reviewer 2: 

Thank you for the comments. The caiman name was corrected.

Reviewer 3 Report

Comments and Suggestions for Authors

This is a very interesting piece of historical ethnobiology. It is written with very good language. It contains unique historical discovery worth sharing to the international community of ethnobiologists.

I sugessts quoting a few more papers containing ethnobiological observations  performed during sea voyages, e..g by Alexander von Humboldt, Charles Darwin.

How does the paper fit in the timeline and content of first ethnobotanical explorations of Brazil, and later exploration e.g. by Napoleon Czerniewicz (1812–1881)?

Author Response

Thanks for the quick review and suggestions provided. I accepted most of the suggestions, as explained in the response to each reviewer. All corrections were highlighted, except for the reference numbering, which were all changed from reference number 6 to 66 due to the inclusion of new citations.

Response to Reviewer 3: 

Thank you for the comments and suggestions. I added more well-known authors such as Gandavo, Humboldt, and Darwin to better situate the reader into the timeline of historical registers. However, I opted not to include a comment about the relationship with Czerniewicz’s works because there is no direct linkage with the main issue of this article (otherwise, I should connect the discussions of the article also to other ethnobotanical explorations, which is not the main objective of this paper).